# Collaborative Modality Fusion for Mitigating Language Bias in Visual Question Answering

**DOI:** 10.3390/jimaging10030056

**Published:** 2024-02-23

**Authors:** Qiwen Lu, Shengbo Chen, Xiaoke Zhu

**Affiliations:** School of Computer and Information Engineering, Henan University, Kaifeng 475001, China; 104753210894@henu.edu.cn (Q.L.); whuzxk@whu.edu.cn (X.Z.)

**Keywords:** visual question answering, collaborative learning, language bias

## Abstract

Language bias stands as a noteworthy concern in visual question answering (VQA), wherein models tend to rely on spurious correlations between questions and answers for prediction. This prevents the models from effectively generalizing, leading to a decrease in performance. In order to address this bias, we propose a novel modality fusion collaborative de-biasing algorithm (CoD). In our approach, bias is considered as the model’s neglect of information from a particular modality during prediction. We employ a collaborative training approach to facilitate mutual modeling between different modalities, achieving efficient feature fusion and enabling the model to fully leverage multimodal knowledge for prediction. Our experiments on various datasets, including VQA-CP v2, VQA v2, and VQA-VS, using different validation strategies, demonstrate the effectiveness of our approach. Notably, employing a basic baseline model resulted in an accuracy of 60.14% on VQA-CP v2.

## 1. Introduction

Visual-language tasks typically necessitate models to comprehend features from different modalities for knowledge reasoning. In practical applications such as intelligent service robotics, visual-language tasks play a crucial role [1,2]. Visual question answering (VQA), as one of the tasks within visual-language understanding, aims to answer textual questions based on provided images. An ideal VQA model should possess the ability to comprehend and reason with image-textual data.

However, recent research [3] indicates that many VQA methods tend to rely on superficial correlations between questions and answers, neglecting to extract accurate visual information from images to answer questions. As illustrated in Figure 1, there is often a notable discrepancy in the distribution of answers between the training and test sets within the VQA dataset. Furthermore, as highlighted in [4], similar bias issues are influenced by visual modalities.

These biases, prevalent in the current VQA domain, mainly involve inherent language distribution biases in training and test sets, as well as incorrect visual grounding shortcuts due to significant visual regions [5,6].

Currently, prominent techniques to address biases involve integration-based [3], counterfactual-based [7,8], and contrast learning-based [9] methodologies. Notably, the integration-based approach lessens bias effects by training two models comprehensively, where one of them captures shallow or spurious associations, allowing the main model to concentrate on more complex instances. Counterfactual-based techniques support training by producing counterfactual samples and supplementary visual annotations. Contrastive learning-based techniques amplify problem contribution by generating negative sample image-problem pairs with irrelevant images from the training data.

However, certain studies [10] have observed that the effectiveness improvement of some methods is not due to a reasonable visual basis but rather an undisclosed regularization effect. Current methods focus on modeling dataset biases to mitigate their influence during de-biasing but overlook the model’s ability for modal understanding and inference. Therefore, our approach aims to enhance the model’s ability to comprehend multimodal information. We incorporate collaborative learning into multimodal training [11] to address bias issues in VQA and reduce its impact.

We classify VQA’s bias problem as a scenario where two parallel modalities are involved in training, but one modality is absent or fails to fulfill its intended function. For instance, the question-answer shortcut bias refers to the model relying solely on the shortcut between the question and the answer for answering, disregarding the relevant visual region (visual bias refers to the same phenomenon). Prior research has demonstrated that this methodology entirely omits visual modal information during the reasoning process. Despite modal feature fusion, the model still disregards the image content for the final prediction and, instead, relies on the bias to answer the question.

Inspired by the concept of collaborative learning, we present an intuitive multimodal training approach to enhance the model’s comprehension of visual text characteristics. Our approach entails leveraging various modalities to reinforce one another during training, thereby mutually aiding the training process. As illustrated in Figure 2, conventional VQA methods are susceptible to verbal and visual biases during training. When confronted with biased questions, these methods usually answer the questions directly based on the influence of bias. In the “CoD-VQA” approach, the model initially identifies possible bias in the present example and its type. A co-operative learning methodology integrates both modalities equally to support each other’s training. This implementation enables the model to use multimodal knowledge thoroughly to make predictions and reduce bias.

In our experiments, we categorize single visual and verbal modal information to enable the model to make predictions. We then compare the unimodal prediction results with real results to identify missing modalities in co-operative learning. Finally, we re-represent and integrate these missing modalities to alleviate bias and enhance participation in prediction.

Overall, we propose a co-training de-biasing method. In our approach, the issue of bias resulting from visual and textual modalities is viewed as a common underlying problem, leading us to suggest a new fusion technique that tackles this problem by focusing on the modal characteristics and adopting the principle of collaborative learning. During the training phase, we assess the parallel visual textual modal data, identifying the “impoverished” and “enriched” modalities. We augment the role of the “deprived” modality artificially to increase the model’s awareness of its presence and reduce bias.

## 2. Related Work

### 2.1. Language Bias

The bias problem in VQA has received a great deal of attention, and a lot of excellent work has emerged from ongoing research. Firstly, the most straightforward way to mitigate the bias problem is to construct balanced datasets, e.g.,the auhtors of [12] added complementary scenarios with opposing answers for binary questions in the VQA v2 dataset. Second, the ensemble model-based regularization [3,13,14,15,16,17,18] approach relies on two integrated learning models to deal with linguistic biases to improve the robustness of the main model. Third, adversarial methods [19], such as using adversarial losses, are used to reduce known sources of bias by inducing errors in the model when it is presented with only the question. Fourth, contrast learning-based methods [4,20,21,22] are used to enhance the utilization of information between the visual context and the question by constructing negative image-question pairs. Fifth, an additional annotation-based approach, refs. [23,24] aid training in mitigating linguistic bias through pre-trained human attention maps or additional counterfactual samples. In this paper, we approach multimodal reasoning from the perspective of feature fusion, which makes the model treat the two modalities as an equal “status” and strengthens the role of the deprived modality in training to mitigate the bias problem.

### 2.2. Multimodal Fusion

In multimodal tasks, in addition to dealing with the relationship between data to alleviate the bias problem, improving the model’s ability to understand the data is also one of the ways to deal with the bias problem. Multimodal fusion methods are one of the very important topics in multimodal deep learning, and the existing mainstream fusion methods can be categorized into early fusion, late fusion, and hybrid fusion. The proposed multimodal fusion [25] can provide a more stable prediction for multimodal prediction, and, at the same time, it can supplement the information between multiple modalities in the case of missing modalities. In VQA, multimodal fusion methods can be classified into simple fusion, attention-based fusion, and tensor-based fusion methods, the most important of which is the fusion method based on bi-linear pooling in attention and tensor. The multimodal fusion methods are currently used for better feature representation in VQA and have not been applied to VQA de-biasing. The core idea of multimodal fusion is to obtain more significant feature representations so that the model receives information from different modalities. Therefore, in our approach, we reconsider how multimodal fusion methods can be useful in the VQA de-biasing problem by playing the role of each modality when the modalities are inferencing.

### 2.3. Collaborative Learning

Collaborative learning refers to helping model resource-poor modalities by leveraging knowledge from a resource-rich modality. The classification of modal data defined in collaborative learning can be categorized into parallel, nonparallel, and hybrid data. In parallel data, the authors of [26] used CAA to construct acoustic features on acoustic and articulatory data and construct an unimodal representation of the generated acoustic by using articulatory data. With nonparallel data dominated by transfer learning, the authors of [27] used text to improve image classification by combining CNN visual features with word2vec textual features trained on large-scale datasets.

In this paper, we consider visual textual modalities in VQA as parallel data in determining rich and scarce modalities, using the idea of collaborative learning to help link between modalities, as well as to help the model to understand the modalities, thus alleviating the bias problem.

## 3. Methods

Figure 3 depicts an overview of our CoD-VQA, where we consider the relationship between visual modality, textual modality, language bias, and visual bias to obtain more accurate modal representations and model comprehension.

We specifically examine the impacts of visual bias and language bias separately within the model. During training, we dynamically analyze sample pairs to identify the ‘missing’ modality, aiding the model in acknowledging and understanding the modality better, and we increased the participation of that modality in the model to remove the bias. The approach incorporates a ‘bias detector’ to identify the present bias. Upon determining the bias type, the model identifies the ‘missing’ modality and incorporates it as a ‘third’ modality in the overall modal fusion. During actual training, the fusion process depicted on the right side of Figure 3 will occur only on one side.

### 3.1. Definition of Bias

For the VQA task, conventional methods typically approach it as a multi-class classification problem. The model is focused on the provided triplets $D={\left\{v_{i},q_{i},a_{i}\right\}}_{i=1}^{N}$, in which the *i*-th image $v_{i}\in V$, the question $q_{i}\in Q$, the answer $a_{i}\in A$, primarily aim to train a mapping that accurately allocates responses across the answer set A. When a categorization layer with fixed parameters and only one modality (either visual or textual) is given as input, the model predicts the answer distribution. In our testing, we found that the model maintains a certain level of accuracy when solely provided with either visual or question features, particularly when using UpDn as the baseline model. Alongside insights from [4], we integrated the concept of visual bias into our approach. We defined this concept in Equation (Equation 1):(1)$$B_{v}=c_{v}\left(\sigma \left(v_{i}\right)\right),$$
where $B_{v}$ denotes the distribution of answers with visual bias, $v_{i}$ denotes image $v_{i}\in V$, and $c_{v}$ denotes the only visual classification network. We consider $B_{v}$ as the bias on the image side of the model.

In summary, we consider visual bias in our approach as a complementary aspect to solve the bias problem in VQA and treat it, together with linguistic bias, as a multimodal collaborative de-biasing problem to be solved.

### 3.2. Multimodal Collaborative Learning in VQA

In this section, we focus on the concepts related to collaborative learning.

In multimodal scenarios, especially when modal resources are limited, it becomes crucial to accurately represent modal information as well as the multimodal knowledge required in the model inference process. Collaborative learning aims to utilize the knowledge of a relatively resource-rich modality to assist in modeling a resource-poor modality. Methods based on the concept of collaborative learning can improve the representation performance of not only multimodal but also unimodal data. According to the difference of training resource categories in collaborative learning, collaborative learning methods can be categorized into three types:Parallel data methods: With parallel data, the observation data of one modality in the dataset is required to be directly associated with the observation data of another modality. For example, in a video-audio dataset, the video and voice samples must come from the same speaker.Nonparallel data methods: With nonparallel data, methods do not require a direct correlation between different modalities, and usually, methods in this data context achieve co-learning through overlap at the category level. For example, in OK-VQA, multimodal datasets are combined with out-of-domain knowledge from Wikipedia to improve the generalization of quizzes.Hybrid data approach: With hybrid data, different modalities are connected to each other through shared modalities. For example, in the case of multilingual image captioning, the image modality always matches the caption in any of the languages, and the role of intermediate modalities is to establish correspondences between the different languages so that the images can be associated with different languages.

Overall, collaborative learning aims to utilize complementary information across modalities so that one modality can influence the other, thus creating better multimodal fusion models. In the next section, we will further describe how collaborative learning can be combined with VQA.

### 3.3. CoD-VQA

In VQA, the question-image pairs in the dataset used for training tend to be strongly correlated, i.e., the entity words in the questions tend to have corresponding detection regions in the images, which suggests that there is the same semantics between the images and the questions and that the model can only make correct predictions when the semantics contained in the two are unified. In our approach, we view the bias problem as both visual and language modalities that are independent of each other, and all existing bias problems can be viewed as being caused by the model ignoring the role of a particular modality in the prediction.

In this context, when the semantics in the data is relatively simple, the semantics between multiple modalities can be represented by a single modality, and the model can make a correct answer simply based on the semantics of the single modality. When the semantics in the data need to be combined with both image and text, if the semantics of one modality is lost, even if the model makes a correct answer, it can still be considered that the model does not have the ability to understand the multimodal knowledge. Therefore, we regard image and text as parallel data under collaborative learning, and they are directly related in training, unified through semantics, and assist each other in supplementing information for each other.

For example, under the condition of linguistic bias, we can assume that the model gets the answer directly through the question, ignoring the language of visual modality. In collaborative learning, the textual modality is used as the “rich” modality and the visual modality as the “scarce” modality in this training. Similarly, in the visual shortcut condition, the textual modality can be regarded as the “scarce” modality and the visual modality as the “rich” modality. The CoD-VQA algorithm consists of three steps:Bias prediction: A single-branch prediction of instances Q and V in the training dataset to obtain unimodal predictions $B_{q}$ and $B_{v}$.Modality selection: Based on the $B_{q}$ and $B_{v}$ obtained in the previous step, binary cross-entropy calculation is performed with *L* to obtain the corresponding bias loss in different modalities. Then, according to the size of the loss and the result of the bias detector, we determine which modality in the image and text is the “deprived” modality.Modal fusion: After determining which modality is “deprived”, we fix the “enriched” modality and use modal fusion to get a new modal representation, which enhances the “deprived” modality in the joint representation. We use modal fusion to obtain a new modal representation, which enhances the participation of the “scarcity” modality in the joint representation.

### 3.4. Reducing Bias

In this section, we describe how we apply collaborative learning to the VQA de-biasing problem in the context of Algorithm 1.
**Algorithm 1:** CoD-VQA
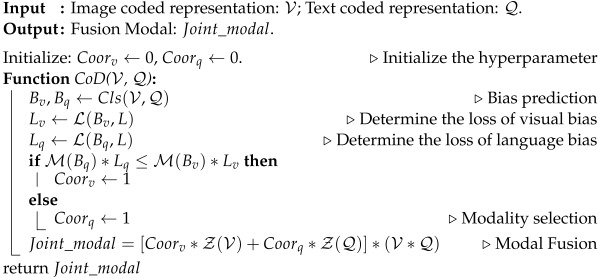


#### 3.4.1. Bias Prediction

Similar to previous research, the most straightforward way to capture VQA bias is to train a model that accepts only one modal input and use it as a bias-capturing branch in the overall model. Specifically, the unimodal bias branch can be represented as Equation (Equation 2):(2)$$\begin{array}{c} \left\{\begin{array}{cc} B_{v} & =c_{v}\left(\sigma \left(v_{i}\right)\right)\\ B_{q} & =c_{q}\left(\sigma \left(q_{i}\right)\right), \end{array}\right. \end{array}$$
where $B_{*}$ (*∈{v,q}) denotes the bias under unimodal branching. $c_{*}$ (*∈{v,q}) denotes the classification prediction layer, which is used to obtain the prediction results. $v_{i}$ denotes image $v_{i}\in V$, and $q_{i}$ denotes question $q_{i}\in Q$.

#### 3.4.2. Selecting the “Scarce” Modality

The key step in our approach is to determine which modality is missing in the bias condition, and we define the missing modality as the “scarce” modality. To be clear, the cause of the bias problem can be explained in terms of missing modalities, which are generated by the model when processing the biased samples; it is not reasonable to identify “scarce” modalities in training by artificial definitions. Therefore, in our approach, we utilize the bias prediction defined in the previous subsection to assist in the judgment. Specifically, after obtaining the biases, we calculate the cross-entropy loss between them and the correct answers and determine which modality should be used as the “scarce” modality based on the size of the resulting loss. The specific process is defined in Algorithm 1 as Equations (Equation 3) and (Equation 4):(3)$$\begin{array}{c} \left\{\begin{array}{cc} L_{v} & =L(B_{v},L)\\ L_{q} & =L(B_{q},L), \end{array}\right. \end{array}$$
(4)$$\begin{array}{c} \left\{\begin{array}{cc} Coor_{v}=1 & \mathrm{if}\;M\left(B_{q}\right)*L_{q}\leq M\left(B_{v}\right)*L_{v}\\ Coor_{q}=1 & \mathrm{if}\;M\left(B_{q}\right)*L_{q}>M\left(B_{v}\right)*L_{v}, \end{array}\right. \end{array}$$
where $L_{*}$ (*∈{v,q}) denotes the loss of the corresponding single-branch bias after cross-entropy computation with the true prediction, respectively, and $Coor_{*}$ (*∈{v,q}) denotes the “scarce” modality identified in the methodology, which has an initial value of 0. mathcalM represents the bias detection classifier, which is used to detect whether the prediction corresponding to the current unimodal mode can be considered as biased. In our approach, we intuitively determine the “rich” modality by comparing the loss corresponding to the bias: a biased pair of samples usually corresponds to a prediction that is initially closer to the true result, which corresponds to a lower loss, whereas the other modality can be considered as a “scarce” modality. However, this approach is based on the assumption that all samples of the training data are biased, whereas in reality, not all samples are biased, or the presence of bias in the samples does not always have a negative impact on training. Therefore, we introduce the M classifier as a bias detector for determining the degree of bias in the current sample.

#### 3.4.3. Modality Fusion

After identifying the “scarce” modality, we perform a re-mapping and fusion of the modalities. Inspired by the work of CF-VQA [15], we consider the bias induced by each single modality and the direct impact on the model to be mutually independent. Consequently, we re-map the features of the “scarce” modality and fuse them with the original modality, represented as Equation (Equation 5):(5)$$Joint\_modal=[Coor_{v}*Z\left(V\right)+Coor_{q}*Z\left(Q\right)]*\left(V*Q\right),$$
where Joint_modal denotes the newly fused mixed modality, and Z represents the mapping layer used for feature handling. The mapping layer employs a conventional fully connected neural network (FCNet) comprising two standard linear layers sequentially stacked.

During the training process, we adopt a two-stage training approach to update the different phases of the algorithm, as illustrated in Figure 4. In the first training phase, the model determines the “scarce” and “rich” modalities during training based on modeled biases and the bias detector, updating the relevant parameters of the bias detector. In the second training phase, based on the identified modalities, we proceed with a new round of modality fusion to ensure the model can recognize and predict from different modality sources, updating the classification layers used for prediction.

## 4. Experiments and Results

In this section, we present the experimental results of our method on different datasets to validate its effectiveness.

### 4.1. Datasets and Evaluation

Our experiments were mainly tested on the VQA v2 and VQA-CP v2 datasets, where the VQA-CP v2 dataset is composed by reorganizing the VQA v2 dataset with the aim of making the answers under the same question word exactly opposite in the training and test sets. In addition, the VQA-VS dataset, which is a new benchmark proposed in a recent study for the bias assessment benchmark problem, is similarly constituted by re-slicing the VQA v2 dataset and setting a shortcut based on different elements as a bias selection. For the evaluation of the results, we use the standard VQA evaluation metrics: answer prediction accuracy and the CGD criterion, which is used to assess whether visual information is employed in answer decision-making.

### 4.2. Results for the VQA-CP v2 and VQA v2 Datasets

#### 4.2.1. Quantitative Results

We compared our CoD-VQA method with state-of-the-art methods on VQA-CP v2 and VQA v2 using the evaluation metric of answer prediction accuracy, as shown in Table 1. From the table, we observe the following:When compared to the backbone models used, our method demonstrates a significant improvement over the UpDn baseline model, achieving an approximate 20% performance boost, showcasing the effectiveness of our approach in reducing language bias.Our method also exhibits considerable performance gains compared to other de-biasing methods. CoD-VQA achieves state-of-the-art performance without using additional annotations, obtaining the best overall accuracy in the “All” category. In specific question types, the CF-VQA variant performs best in “Y/N” questions, while CoD shows better performance in question types requiring more visual content labeled as “Other”.When compared with methods employing data augmentation and additional annotations, our approach similarly demonstrates competitive performance. When using the same UpDn baseline model, our method exhibits an approximate 1.8% improvement over the latest feature-enhancement method, D-VQA. Additionally, CoD outperforms in the “Other” question category, strongly validating the efficacy of our de-biasing approach.On the VQA v2 dataset, our method displays robust generalization capabilities, overcoming the constraints of unknown regularization effects present in the v2 dataset.

#### 4.2.2. Qualitative Results

In order to further demonstrate the effectiveness of our method on the dataset, we present visual cases in Figure 5. The first row illustrates examples of visual shortcut biases. Under the prediction of the UpDn model, it mis-identifies the correct region despite covering a part related to the question, resulting in an incorrect answer. Our model accurately localizes the image region and provides the correct answer. The second row showcases instances of language bias. Both UpDn and our model correctly identify the relevant image regions. However, UpDn tends to utilize distribution bias to answer questions, whereas our model leverages visual information to provide the correct answer. In our analysis, the UpDn model tends to be influenced by biases present within the dataset, often overlooking the inherent connections between data points. In contrast, CoD-VQA manages to recapture these connections to a certain extent, facilitating more effective modality fusion. This distinction underscores the importance of addressing dataset biases and highlights CoD-VQA’s improved handling of modality integration within the context of biased datasets.

### 4.3. Ablation Studies

#### 4.3.1. Modality Selection Evaluation

Within the initial phase of our method’s training process, we established a modality selection module to identify the “scanty modality”. In order to assess the efficacy of this modality, we conduct experiments on the VQA-CP v2 dataset by excluding the modality selection module and employing a fixed “scanty” modality instead. The experimental results are shown in Table 2:

From Table 2, the following conclusions can be drawn:Distinguishing different scanty modalities within samples has a beneficial impact on model performance.Language modality biases are more challenging in the overall bias problem in VQA compared to visual modality biases. When we default the “scanty” modality to the visual modality, the model’s performance improves slightly compared to when fixed as the language modality.

These results indicate that determining the “scanty” modality contributes to enhancing model performance.

#### 4.3.2. Comparison of Other Baseline Models

In this subsection, we examine the effectiveness of our method by utilizing pre-trained models and other baseline models. We test our approach on the VQA-CP v2 dataset, and the results are shown in Table 3.

Table 3 presents the results of our CoD method on the LXMERT pre-trained model. It’s evident that our method exhibits more significant performance improvements when using LXMERT as the baseline model. Our primary objective is to facilitate modalities “aiding” each other in information acquisition, leading to better fusion between different modalities. The modality encoders obtained by large-scale pre-trained models are more effective in representing features for the same samples. Hence, we employ the feature extractor from the LXMERT encoder layer in our method for efficient feature representation. The results indicate some performance gains on top of LXMERT. Additionally, compared to other LXMERT-based D-VQA versions, CoD demonstrates competitive performance. Meanwhile, our approach exhibited a certain degree of performance enhancement when combined with other baseline models.

#### 4.3.3. VQA-VS

Recent studies highlight that the composition of the VQA-CP v2 dataset considers only one type of shortcut, i.e., the shortcut between question types and answers, while VQA-VS reconstructs the VQA v2 dataset, considering three different shortcut categories to include various false associations. Therefore, we conducted experiments for CoD on the VQA-VS benchmark, and the experimental results are shown in Table 4.

In Table 4, we report the performance of CoD-VQA under various shortcuts in the VQA-VS benchmark. In Table 4, various evaluation metrics can be categorized into language modality-based shortcuts: QT (Question Type), KW (Keyword), KWP (Keyword Pair), QTKW (Question Type + Keyword); visual modality-based shortcuts: KO (Key Object), and KOP (Key Object Pair) and cross-modal shortcuts: QTKO (Question Type + Key Object), KWKO (Keyword and Key Object), QTKWKO (Question Type, Keyword, and Key Object).

Our method exhibits competitive performance when addressing specific shortcut issues in the VQA-VS dataset using the same baseline model. In cases of language modality shortcuts, our method notably improves performance regarding biases induced by keywords. Similarly, our approach shows some enhancement in the visual modality aspect. This demonstrates the effectiveness of our method in mitigating biases from a modality perspective.

### 4.4. Analysis of Other Metrics

In our approach, we enable collaborative training among different modalities by identifying the “scarce” modality. In order to validate whether this “scarce” modality is involved in the model’s final reasoning process, we used additional evaluation standards on the VQA-CP v2 dataset to assess the involvement of the “visual modality” in our method.

For a detailed understanding of CGD, please refer to the papers [3,10]. Here, we provide a brief overview. The core idea behind CGD is that during inference, the model’s utilization of visual information should not only contribute to correct predictions based on appropriate visual evidence but also highlight incorrect answers resulting from inappropriate visual cues.

In Table 5, we present a comparison between our method and others in terms of CGD. When compared to GGE [3], our approach reduces the rate of the model using incorrect visual regions to answer questions, thereby achieving a certain level of accuracy in the overall CGD standard, akin to the GGE method. The enhancement in CGD performance indicates an improvement in our method’s ability to leverage visual information for answer prediction.

## 5. Discussion and Conclusions

Visual question answering (VQA) has emerged as a key task within multimodal research, marking a foundational step toward the realization of true artificial intelligence entities. This study explores modal fusion methods in VQA contexts [28,29,30] and suggests that similar approaches could be beneficial for other multimodal tasks, such as image captioning, especially in identifying biases. Long-tail distributions in answer datasets and biases due to missing modal information in images represent unavoidable challenges in VQA development. Unlike prior work, this study addresses a fundamental issue in multimodal tasks: the model’s comprehension across different modalities, highlighting the necessity of overcoming dataset limitations and bias to fully capture multimodal interactions.

In this paper, we have introduced a de-biasing model for VQA based on multimodal collaborative training. Our approach considers image and text features in VQA as equally important modalities and employs the concept of collaborative learning to assist each other in training, mitigating bias issues from a modality feature perspective. Specifically, within the de-biasing process, we defined symmetrical language and visual biases, categorizing the reasons behind biases as attentional deficits of modality information during model predictions. Subsequently, we further utilized the concept of collaborative learning to define the missing “scarce” modality during training. By leveraging mutual assistance among modalities in training, we aimed to achieve better modal fusion and feature representation, thereby addressing bias issues. Our extensive experiments conducted on benchmark datasets, VQA-CP v2 and VQA v2, and the novel de-biased dataset VQA-VS, demonstrate the effectiveness of our CoD-VQA method in tackling bias-related problems.

In summary, we have developed a multimodal collaborative de-biasing algorithm that, while adopting a modal fusion approach to bias mitigation, still faces certain limitations. Primarily, the dataset on which the algorithm is based does not encompass all real-world scenarios, leading to challenges in generalizing to “unusual” questions in practical contexts. Moreover, the algorithm’s effectiveness hinges on precise bias detection and the modality fusion’s performance under biased conditions. Given the complexity and variability of real-world scenarios, the model may not capture these nuances effectively. Future work could focus on enhancing dynamic bias detection and modal fusion techniques to ensure broader robustness in VQA applications.

## Figures and Tables

**Figure 1 jimaging-10-00056-f001:**
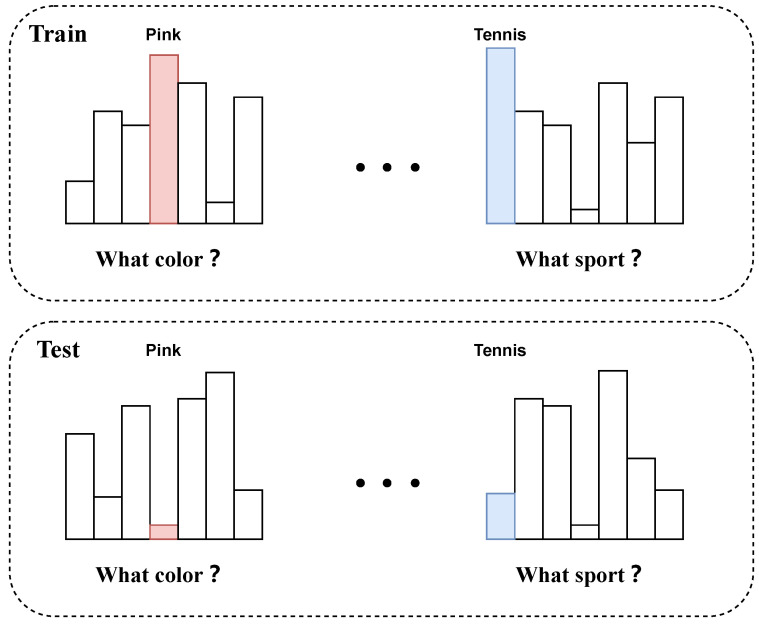
Distribution bias: the answer distribution of the training set and test set is significantly different.

**Figure 2 jimaging-10-00056-f002:**
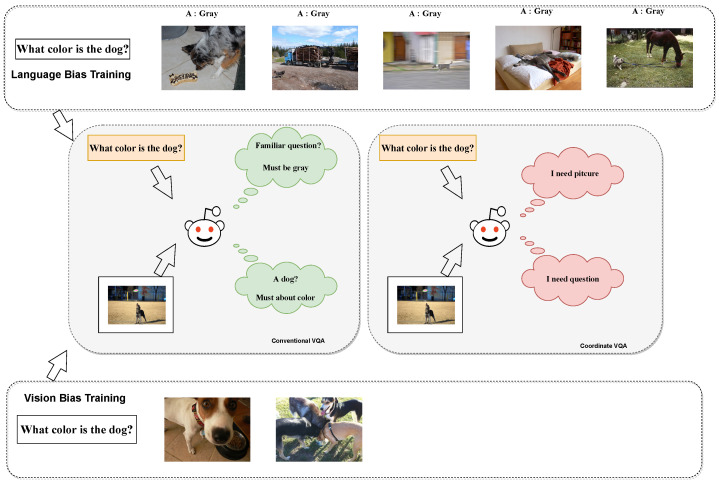
Existing VQA models are impacted by biases originating from both linguistic and visual domains. Our model forcibly discerns information from diverse modalities during training, maximizing data utilization for predictions.

**Figure 3 jimaging-10-00056-f003:**
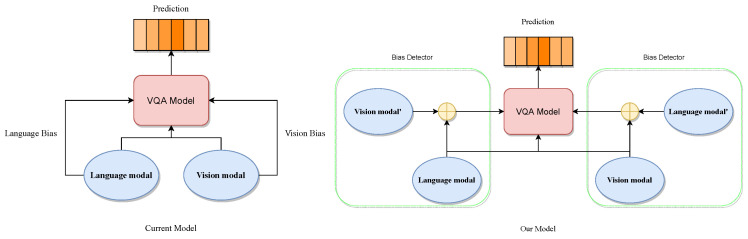
In the comparison between our method and previous VQA models, the Bias Detector signifies a bias detector used to identify the type of bias present in the current sample. In our model, the modality fusion process on the left and right sides only occurs unilaterally during a single training iteration. The depicted blue ellipse symbolizes the module responsible for extracting features from visual and textual data, whereas the red rectangle denotes the module dedicated to fusing these modalities. Furthermore, within the Prediction segment, the varying intensities of color serve to illustrate the differential weighting attributed to the outcomes predicted by the model.

**Figure 4 jimaging-10-00056-f004:**
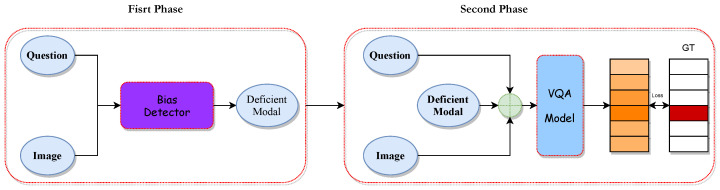
Two-stage training process in our approach: the first stage focuses on bias identification and modality determination, while the second stage re-fuses the “scarce” modality and predicts answers. The blue elements symbolize the modal features, whereas the purple rectangles are indicative of bias detectors, which are used to identify missing modes in a round of training data.

**Figure 5 jimaging-10-00056-f005:**
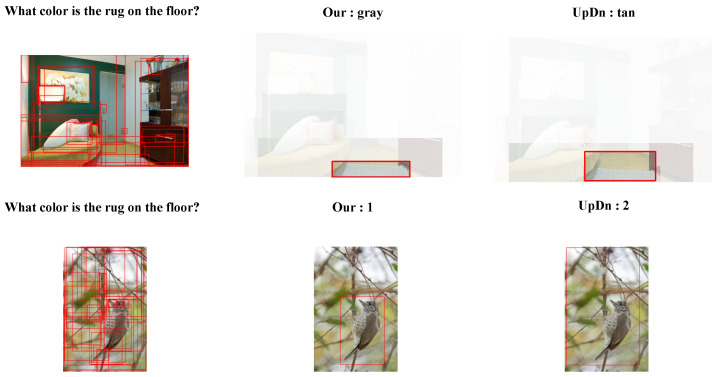
Qualitative comparison of our CoD-VQA with UpDn on the VQA-CP v2 test set, focusing on language and visual biases. The red rectangle box represents the output of the image after target detection and is used to represent the detected object in the image. The red wire boxes in the middle and on the right indicate the objects corresponding to the predicted answers.

**Table 1 jimaging-10-00056-t001:** The results of VQA-CP v2 test set and VQA v2 validation set are presented in the following table. Each column illustrates the **best** performances of each method, excluding data augmentation techniques. Our CoD method has been compared with state-of-the-art methods on both datasets.

Method	Base	VQA-CP v2 Test				VQA v2 Val			
		All	Y/N	Num.	Other	All	Y/N	Num.	Other
GVQA	-	31.30	57.99	13.68	22.14	48.24	72.03	31.17	34.65
SAN	-	24.96	38.35	11.14	21.74	52.41	70.06	39.28	47.84
UpDn	-	39.96	43.01	12.07	45.82	63.48	81.18	42.14	**55.66**
S-MRL	-	38.46	42.85	12.81	43.20	63.10	-	-	-
HINT	UpDn	46.73	67.27	10.61	45.88	63.38	81.18	42.99	55.56
SCR	UpDn	49.45	72.36	10.93	48.02	62.2	78.8	41.6	54.5
AdvReg	UpDn	41.17	65.49	15.48	35.48	62.75	79.84	42.35	55.16
RUBi	UpDn	44.23	67.05	17.48	39.61	-	-	-	-
RUBi	S-MRL	47.11	68.65	20.28	43.18	61.16	-	-	-
LM	UpDn	48.78	72.78	14.61	45.58	63.26	81.16	42.22	55.22
LMH	UpDn	52.01	72.58	31.12	46.97	56.35	65.06	37.63	54.69
DLP	UpDn	48.87	70.99	18.72	45.57	57.96	76.82	39.33	48.54
DLR	UpDn	48.87	70.99	18.72	45.57	57.96	76.82	39.33	48.54
AttAlign	UpDn	39.37	43.02	11.89	45.00	63.24	80.99	42.55	55.22
CF-VQA(SUM)	UpDn	53.55	**91.15**	13.03	44.97	**63.54**	**82.51**	**43.96**	54.30
Removing Bias	LMH	54.55	74.03	49.16	45.82	-	-	-	-
CF-VQA(SUM)	S-MRL	55.05	90.61	21.50	45.61	60.94	81.13	43.86	50.11
LPF	UpDn	55.34	88.61	23.78	46.57	55.01	64.87	37.45	52.08
GGE-DQ-iter	UpDn	57.12	87.35	26.16	49.77	59.30	73.63	40.30	54.29
GGE-DQ-tog	UpDn	57.32	87.04	27.75	49.59	59.11	73.27	39.99	54.39
GGD	UpDn	59.37	88.23	38.11	49.82	62.15	79.25	42.43	54.66
AdaVQA	UpDn	54.67	72.47	**53.81**	45.58	-	-	-	-
ECD	LMH	59.92	83.23	52.59	49.71	-	-	-	-
CoD(Ours)	UpDn	**60.14**	85.66	39.08	**52.54**	62.86	78.65	45.01	54.13
**Method**	**Base**	**VQA-CP v2 Test**				**VQA v2 Val**			
		**All**	**Y/N**	**Num.**	**Other**	**All**	**Y/N**	**Num.**	**Other**
*Methods of data augmentation and additional annotation:*									
AttReg	LMH	59.92	87.28	52.39	47.65	62.74	79.71	41.68	55.42
CSS	UpDn	58.95	84.37	49.42	48.24	59.91	7.25	39.77	55.11
CSS+CL	UpDn	59.18	86.99	49.89	47.16	57.29	67.27	38.40	54.71
Mutant	UpDn	61.72	88.90	49.68	50.78	62.56	82.07	42.52	53.28
D-VQA	UpDn	61.91	88.93	52.32	50.39	64.96	82.18	44.05	57.54
KDDAug	UpDn	60.24	86.13	55.08	48.08	62.86	80.55	41.05	55.18
OLP	UpDn	57.59	86.53	29.87	50.03	-	-	-	-
SAR	LMH	66.73	86.00	62.34	57.84	69.22	87.46	51.20	60.12

**Table 2 jimaging-10-00056-t002:** Effect of different “scanty” modalities on model accuracy. indicates that a mode is fixed as “scanty” in the experiment.

	Language	Vision	Both	VQA-CP Test
1	✓			54.72
2		✓		56.36
3			✓	60.14

**Table 3 jimaging-10-00056-t003:** **CoD-VQA** experimental results on pre-trained models and performance on different baseline models.

Model	Yes/No	Num.	Other	Overall
SAN	39.44	12.91	46.65	39.11
+CoD	81.81	47.46	38.20	52.56
UpDn	43.01	12.07	45.82	39.96
+CoD	85.66	39.08	52.54	60.14
RuBi	67.05	17.48	39.61	44.23
+CoD	79.93	45.78	46.04	55.87
LXMERT	42.84	18.91	55.51	46.23
+CoD	82.51	57.84	58.64	65.47
+D-VQA	80.43	58.57	67.23	69.75

**Table 4 jimaging-10-00056-t004:** Experimental results of CoD-VQA on the VQA-VS dataset, showcasing the relevant experimental report findings from this dataset. Each column displays the corresponding **best** and second-best performance results.

		VQA-VS OOD Test Sets									
**Model**	**Base**	**Language-Based**				**Visual-Based**		**Multi-Modality**			**Mean**
		**QT**	**KW**	**KWP**	**QT + KW**	**KO**	**KOP**	**QT + KO**	**KW + KO**	**QT + KW + KO**	
S-MRL	-	27.33	39.80	53.03	51.96	27.74	35.55	42.17	50.79	55.47	42.65
UpDn		32.43	45.10	56.06	55.29	33.39	41.31	46.45	54.29	56.92	46.80
+LMH	UpDn	33.36	43.97	54.76	53.23	33.72	41.39	46.15	51.14	54.97	45.85
+SSL	UpDn	31.41	43.97	54.74	53.81	32.45	40.41	45.53	52.89	55.42	45.62
BAN	-	33.75	46.64	58.36	57.11	34.56	42.45	47.92	56.26	59.77	48.53
LXMERT	-	**36.46**	**51.95**	**64.17**	**64.22**	**37.69**	**46.40**	**53.54**	**62.46**	**67.44**	**53.70**
CoD-VQA (Ours)	UpDn	32.91	49.65	62.65	61.51	34.46	43.58	51.47	60.84	66.35	51.49

**Table 5 jimaging-10-00056-t005:** Experiment on the evaluation metric CGD using the CoD-VQA method on the VQA-CP v2 dataset. The **best** results are displayed in each column.

Method	CGR	CGW	CGD
UpDn	44.27	40.63	3.91
HINT	45.21	34.87	10.34
RUBi	39.60	33.33	6.27
LMH	46.44	35.84	10.60
CSS	**46.70**	37.89	8.87
GGE-DQ-iter	44.35	27.91	**16.44**
GGE-DQ-tog	42.74	27.47	15.27
CoD (Ours)	37.50	**21.46**	16.04

## Data Availability

Publicly available datasets were analyzed in this study. All the data used in this paper are available in these collections: https://phoebussi.github.io/VQA-VS-homepage/ (accessed on 18 February 2024); https://www.iro.umontreal.ca/~agrawal/vqa-cp/ (accessed on 18 February 2024); https://visualqa.org/index.html (accessed on 18 February 2024). Our code can be read at: https://github.com/EASONGLLL/Coor (accessed on 18 February 2024).

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
