# Peer review of "Collaborative Modality Fusion for Mitigating Language Bias in Visual Question Answering"

_2313-433X, 2024, doi:10.3390/jimaging10030056_

Round 1

Reviewer 1 Report

Comments and Suggestions for Authors

The paper demonstrates a robust understanding of the technical aspects of Language bias stands and its application in Visual Question Answering (VQA). The article is very interesting and suitable for any reader in this field, but some recommendations should be considered for publication:

Comments:

1.      The manuscript structure should include an Abstract, Keywords, Introduction, Materials and Methods, Results, Discussion, and Conclusions. The sections Results and Discussion should be created.

2.      Authors should discuss the results and how they can be interpreted from the perspective of previous studies.

3.      In this reviewer´s view, a brief comment on the future Language bias stands and VQA is necessary in the conclusions.

4.      References should be numbered in order of appearance and indicated by a numeral or numerals in square brackets—e.g., [1] or [2,3], or [4–6].

5.      Don`t write in the first-person point of view, for example “we understand…” (line 427) the correct way is with impersonal constructions.

6.      Check references, there are mistakes. They do not adapt to the MDPI format. For example, the correct format of paper is: Author 1, A.B.; Author 2, C.D. Title of the article. Abbreviated Journal Name Year, Volume, page range.

7.      Include more scientific references about the topic, for example:

J. Zhao, Z. Yu, X. Zhang and Y. Yang, "Overcoming Language Priors via Shuffling Language Bias for Robust Visual Question Answering," in IEEE Access, vol. 11, pp. 85980-85989, 2023, doi: 10.1109/ACCESS.2023.3304415.

Yuan, D.; Wang, L.; Wu, Q.; Meng, F.; Ngan, K.N.; Xu, L. Language Bias-Driven Self-Knowledge Distillation with Generalization Uncertainty for Reducing Language Bias in Visual Question Answering. Appl. Sci. 202212, 7588. https://doi.org/10.3390/app12157588

1. The authors propose a new collaborative modality fusion bias elimination (CoD) algorithm. They employ a collaborative training approach to facilitate mutual modelling between different modalities.
2-3. I consider original the introduction in the implementation of a de-biasing model for Visual Question Answering (VQA) based on multimodal collaborative training.
2. I do not consider that there is a gap in this area, I only propose changes to improve the quality of the manuscript.
4-5. I believe that the methodology and conclusions are correct and provide an understanding of all the main issues.
5. In this reviewer´s view, a brief comment on the future Language bias stands and VQA is necessary in the conclusions.
6-7. References, tables and figures are appropriate.

Author Response

Thank you very much, Reviewer 1, for taking the time to provide feedback on our manuscript. We have carefully reviewed your comments and made the necessary modifications and responses in the attached document. Please take a look at them. Your professional insights are greatly appreciated. If you have any further questions or areas for discussion regarding the modifications, please feel free to let us know.

Reviewer 2 Report

Comments and Suggestions for Authors

This research article introduces a de-biasing model for Visual Question Answering (VQA) based on multimodal collaborative training, addressing the significant issue of language bias in VQA systems. The authors propose a novel algorithm, Collaborative Modality Fusion (CoD-VQA), to mitigate language bias by considering the neglect of information from a particular modality during prediction and using collaborative training to facilitate mutual modeling between different modalities.

Strengths:

  1. By focusing on collaborative modality fusion, the CoD-VQA algorithm introduces a fresh perspective compared to existing techniques that address language bias in VQA systems.

  2. The authors thoroughly analyze the impact of visual and language biases separately, incorporating a bias detector and focusing on 'missing' modalities to increase their participation in model predictions.

  3. The article presents robust experimental validation on multiple datasets (VQA-CP v2, VQA v2, VQA-VS) using different validation strategies, demonstrating the effectiveness of the proposed approach with notable accuracy improvements.

  4. The methodology section clearly outlines the three steps of the CoD-VQA algorithm: bias prediction, modality selection, and modal fusion, providing clarity on the working of the proposed model.

  5. Relevance and Timeliness: Addressing bias in AI systems, especially in multimodal contexts, is highly relevant and timely given the increasing reliance on AI in various sectors.

Suggested areas of improvement:

  1. Broader Contextualization: Although the focus of this paper is on VQA, it is possible to enhance the relevance of this work if it is discussed in a broader context regarding the implications of this work in other multimodal AI systems.

  2. Improved Comparative Analysis: A thorough comparison of the CoD-VQA method with existing de-biasing methods could highlight its unique contributions and advantages.

  3. Incorporating a user study or real-world application scenario could provide valuable insights into how the CoD-VQA model performs in practical settings.

  4. The paper would benefit from a more detailed discussion of the limitations of the proposed approach and potential challenges in real-world implementations.

  5. Suggesting future research directions based on the findings and potential extensions of the CoD-VQA model would be beneficial.

Conclusion:

Lu, Chen, and Zhu's article "Collaborative Modality Fusion for Mitigating Language Bias in Visual Question Answering" contributes significantly to the field of VQA and AI bias mitigation. In addition to its innovative approach, comprehensive analysis, and robust experimental validation, this work represents a notable advance in the field. The paper holds promising potential for future developments in unbiased AI systems when enriched with broader contextualization, comprehensive comparative analysis, and discussion of practical applications and limitations.

Author Response

Thank you very much, Reviewer 2, for taking the time to provide feedback on our manuscript. We have carefully reviewed your comments and made the necessary modifications and responses in the attached document. Please take a look at them. Your professional insights are greatly appreciated. If you have any further questions or areas for discussion regarding the modifications, please feel free to let us know.

Reviewer 3 Report

Comments and Suggestions for Authors

The paper is written in an easy-to-read manner and clearly demonstrates the contribution of the proposed algorithm.

1) Please adjust the references in the paper so that they are cited in order. For example, please adjust in the following order: [1][2][3].

2) The number of references is relatively small. In particular, the number of papers published recently at major conferences is small. I hope that more CoD-related studies will be added. [12][13] [14] is a paper published in 2023, and [21] is a paper published in 2022, but since it is a journal paper, it is insufficient to say that it reflects recent research trends.

3) In Algorithm 1, Z(V) Z(Q)?:

Z represents the mapping layer used for feature handling. Specifically, how are the layers structured?

Also, add a brief comment for each line in Algorithm 1

4) Please revise the comparative experiment to ensure that comparative experiments with the papers below are added.

Q. Si, Z. Lin, M. Zheng, P. Fu, and W. Wang, “Check it again: Progressive visual question answering via visual implications,” arXiv preprint arXiv:2106.04605, 2021.

HAN, X., WANG, S., SU, C., HUANG, Q., AND TIAN, Q. General greedy de-bias learning. IEEE Transactions on Pattern Analysis 415 and Machine Intelligence (2023).

Zujie Liang, LPF: A Language-Prior Feedback Objective Function for De-biased Visual Question Answering, SIGIR ’21

5) Please add a diagram for 'The answer distributions on the VQA-CP v2' using the proposed method.

Author Response

Thank you very much, Reviewer 3, for taking the time to provide feedback on our manuscript. We have carefully reviewed your comments and made the necessary modifications and responses in the attached document. Please take a look at them. Your professional insights are greatly appreciated. If you have any further questions or areas for discussion regarding the modifications, please feel free to let us know.

Round 2

Reviewer 2 Report

Comments and Suggestions for Authors

No additional comments or suggestions. 

Reviewer 3 Report

Comments and Suggestions for Authors

Although the author's response said that the latest related studies were added, it is difficult to confirm which papers were added in the actual revised paper. I hope the author adds papers published within the last 1 to 3 years to the related research.